# Statistics in the Next Quarter-Century:
# Playing also in the Frontyard?*

**Abstract**

Statistics has undergone remarkable development over the past decades while interacting increasingly with Data Science, Machine Learning, and Artificial Intelligence (AI). This perspective discusses the current position of Statistics, its strengths, and the challenges it faces in a rapidly evolving scientific and technological landscape. I argue that Statistics remains a foundational discipline: rooted in applications and committed to reliability, replicability, and principled reasoning under uncertainty. The field should engage with modern tools from AI that can assist in knowledge discovery and technical work. At the same time, this makes it even more important to cultivate human abilities in problem framing and critical statistical thinking across research, development, and education. With these strengths, Statistics should engage not only in the "backyard" but also in the "frontyard" of formulating important questions and shaping scientific and societal progress.

**Keywords:** Artificial Intelligence, Machine Learning, Reliability of inference, Replicability and Reproducibility

**Mathematics Subject Classification (2020):** 62A01

## 1 Introduction

Is Statistics well prepared for the second quarter of the twenty-first century? Reflecting on the developments of the past twenty-five years provides a useful perspective for considering the challenges and opportunities ahead.

During this period, Statistics has expanded its influence across science, engineering, and industry. At the same time, it has become increasingly intertwined with neighboring disciplines such as Machine Learning, Data Science, and AI. These interactions have produced remarkable intellectual cross-fertilization, but they have also raised questions about disciplinary identity, visibility, and long-term direction.

This article discusses three themes. First, I consider the evolving position of Statistics within the broader Data Sciences. Second, I highlight several conceptual developments from the past 25 years that illustrate the continuing vitality of the field. Finally, I reflect on broader aspects of creativity and style in science, emphasizing that in an era of increasingly automated technical

---

*Instead of only "in the backyard" (Tukey)

work, the human capacities for problem framing, simplicity, conceptual clarity, and elegance will remain central to statistical thinking.

## 2 The position of Statistics in the Data Science ecosystem

### 2.1 Growth of data-oriented professions

Labor market projections provide one lens through which to view the evolving role of Statistics. Reports from the U.S. Bureau of Labor Statistics and other organizations consistently rank Data Science and AI-related occupations among the fastest growing professions, see Figure 1. Data Science is ranked fourth among the fastest-growing occupations, number 3 is "Nurse

| | | |
|---|---|---|
| **Data Scientists** | 34% to 36% | Much faster than average |
| **Computer Research Scientists** | 20% | Much faster than average |
| **Software Developers** | 17% to 18% | Much faster than average |
| **Epidemiologists** | 16% to 19% | Much faster than average |
| **Medical Scientists** | 9% to 11% | Much faster than average |
| **Statisticians** | 8% to 11% | Much faster than average |
| **Biochemists/Biophysicists** | 6% to 9% | Faster than average |

Figure 1: Projected growth rates 2024-2034 from the U.S. Bureau of Labor Statistics. Field (left), growth rate over 10 years (middle), magnitude of growth rate (right). The World Economic Forum predicts 40+% for AI & Machine Learning Specialists, Fortune Business Insights gives 36.2% for ML Engineers. The average (total economy's growth rate) is 3%.

Practitioners" (40%) and number 1 and 2 are professions in renewable energy but with a small number of employees. Thus, clearly, Data Science grows exceptionally fast while the fields of Machine Learning and AI specialists have an even higher projected growth.

These figures also raise an important question: to what extent do such categories reflect substantive conceptual differences between Statistics and Data Science? Within academia, many departments now carry titles such as "Statistics and Data Science," suggesting considerable overlap. In industry and the labor market, however, the labels often signal differentiation. Whether this differentiation reflects genuine methodological distinctions or largely matters of branding and perception remains unclear. Nevertheless, when Statistics is viewed broadly as encompassing modern data analysis, its prospects appear strong. Statistics has maintained steady institutional support and intellectual momentum.[1]

### 2.2 Conceptual innovations over the last quarter of century

Before turning to the challenges and opportunities of the next quarter century, it is worth reflecting about the past twenty-five years, during which Statistics has witnessed a series of major conceptual advances. The examples discussed below are necessarily selective, but they illustrate the breadth and impact of modern statistical research. The first four share a common feature: they address problems that were once considered nearly intractable. The remaining contributions are also among the most influential developments of the period.

*False Discovery Rate (FDR) and Selective Inference.* The False Discovery Rate and its controlling procedure marked a breakthrough in large-scale multiple testing, with profound impact across scientific disciplines (Benjamini

---

[1]Statistics has so far not experienced an analogue of the "AI winters" that occurred in earlier decades.

and Hochberg, 1995). Closely related developments in selective inference address the challenge of valid inference after model selection, sometimes described as "using the data twice." It is striking that rigorous solutions to this problem exist, though not in a naive form. These contributions originated largely within Statistics and have become indispensable tools in many fields outside Statistics, in particular the reporting of the False Discovery Rate.

*High-dimensional models and sparsity.* In the twentieth century, problems with far more parameters than observations ($p \gg n$) were generally regarded as ill-posed. While the case $p = O(n)$ was understood, e.g. under smoothness assumptions, the introduction of sparsity fundamentally changed the landscape (Donoho and Johnstone, 1994; Donoho, 2006). The field developed through contributions from applied mathematics, signal processing, and Statistics, with statisticians playing a central role in methodology, theory, and practical implementation through user-friendly software (Friedman et al., 2010).

*Conformal Inference.* Conformal prediction provides valid predictive inference based on arbitrarily complex and potentially overfitted machine learning algorithms. Conceptually, it is striking that the construction can rely on estimators that may themselves be inconsistent. Although the foundational ideas originated outside statistics (Gammerman et al., 1998), the area has since flourished through substantial contributions from both statistics and machine learning.

*Causality.* Distinguishing cause from effect based on observational data was long considered fundamentally ill-posed. Over the past decades, however, the development of graphical models, structural causal frameworks, and formal identification theory has transformed the study of causality. While many foundational ideas originated outside Statistics in the field of causality (in contrast to causal inference) (Pearl, 2009; Spirtes et al., 2000), several pioneering contributions also emerged from Statistics (Cowell et al., 1999; Dawid, 2000). The area is now strongly cross-fertilized by multiple scientific communities.

*Statistical Machine Learning, and beyond.* "The Elements of Statistical Learning" (Hastie et al., 2001) has shaped an entire generation of statisticians and beyond, including e.g. bioinformatics, econometrics, social sciences and many more. Also Breiman's work exemplifies a statistician shaping Machine Learning. His "two cultures" paper (Breiman, 2001b) challenged conventional statistical modeling, and Random Forests (Breiman, 2001a) became one of the most influential algorithms in modern data analysis.[2] This illustrates that Statistics had branches with impact far beyond disciplinary boundaries.

*Normalization and data processing in genomics.* The extraction of meaningful biological signal from noisy genomic measurements, grounded in statistical principles, has become a central component of modern genomics pipelines. Irizarry et al. (2003) developed an early and influential approach for normalization and data summarization of Affymetrix GeneChip arrays, providing a principled foundation for preprocessing and downstream analysis that shaped subsequent generations of measurement technologies. Later, Love et al. (2014) introduced statistical methods for the analysis of comparative sequencing assays, in particular RNA-seq read counts, together with the accompanying `DESeq2` software package. Both the methodological framework and the software have had enormous impact on genomics research and are now standard tools for the analysis of bulk and single-cell RNA-sequencing data.

*Strengthening scientific practice.* The PRISMA statement from Altman and colleagues (Moher et al., 2009) is to help improve the reporting of systematic reviews and meta-analyses, usually in a medical or clinical context. [3] This is an important example demonstrating that Statistics contributes not only through algorithms competing with Machine Learning and AI, but also through strengthening scientific practice outside the field.

*The R project.* One of the most impactful developments is the R project (R Core Team, 2021). It is worth noting that this achievement emerged from the collective effort of a large community of contributor: something that would not have been possible for a small group alone, and that bears some resemblance to large collaborative initiatives in modern science. Rooted in open-source principles, R has played a pioneering role in promoting reproducible data analysis and has profoundly shaped modern data science, far beyond the field of Statistics.

---

[2]176'807 citations from Google Scholar (Feb. 24, 2026).

[3]192'720 citations from Google Scholar (Feb. 24, 2026); higher than the top-cited deep learning papers, i.e., "Imagenet classification (Krizhevsky et al., 2012)" (190'807 Google Scholar citations) and "Deep Learning (LeCun et al., 2015)" (107'690 Google Scholar citations).

# 3 Directions for the future

Predicting the next wave of statistical innovation would contradict the very nature of surprising innovation itself: the more transformative a breakthrough, the less predictable it tends to be. The developments in Statistics over the past 25 years, outlined above, nevertheless provide some guidance for the future. Statistics should remain a broad discipline that continues to expand in multiple directions. It is rooted in applications, often grounded in mathematics, and committed to reliability and replicability. While individual researchers or groups may not embody all of these aspects, the discipline as a whole should strive to uphold a high standard of credibility.

## 3.1 Statistics: in the backyard as a helping science?

Within Data Science and AI, statisticians often emphasize that statistical ideas are essential for reliable methodology, uncertainty quantification, and principled inference. This claim is well justified. Yet it is less common for machine learning researchers to describe their work as fundamentally contributing to Statistics. As a result, Statistics is sometimes perceived as a "helping science": a discipline that strengthens and validates developments initiated elsewhere. This perception is not new. Much statistical innovation has historically been driven by applications, where methodological advances enabled scientific insights that would otherwise have been unattainable.

However, recent large-scale AI initiatives sometimes place non-domain experts at the center of scientific breakthroughs. Well-known examples include AlphaFold2 with the corresponding 2024 Nobel Prize in Chemistry and half of the prize going to two people at Google Deep Mind, or the Aurora foundation model for weather and environmental forecasting, led by Microsoft Research. And thus, AI researchers have become visible co-leaders in scientific progress.

John Tukey famously remarked that "The best thing about being a statistician is that you get to play in everyone's backyard." This metaphor captures the interdisciplinary nature of Statistics. To play in someone else's backyard, however, one must first enter the house and understand the science within. Genuine collaboration requires immersion in the domain. It is a beautiful picture about the strong tradition of Statistics: a most successful contribution happens through true collaboration. Yet Tukey's metaphor also implies that statisticians are in the backyard, not the front yard. Can Statistics move more visibly into the frontyard without losing its collaborative culture?

There may be a trade-off between prominence and deep interdisciplinarity.[4] Nonetheless, I believe this question deserves careful reflection, and I offer my own perspective below.

### 3.1.1 Statistics in the frontyard

The same characteristic that enables Statistics to contribute effectively in the "backyard", namely its high degree of interdisciplinarity, also enables it to play a central role in the "frontyard". It is important that the discipline is visibly present there as well, not in isolation, but as a co-owner of "the house and the garden". My non-exhaustive list of examples in which Statistics occupies co-leading roles in major interdisciplinary endeavors includes:

*Large clinical trials*, often conducted in collaboration with pharmaceutical companies;

*UK Biobank genotyping*, the *Wellcome Trust Case Control Consortium*, and related large-scale platforms;

*The Human Flourishing Program* and quantitative social science;

*Leadership roles in interdisciplinary teams* driving advances in online experimentation and marketplace design, climate science, weather forecasting and air quality, computational and systems biology, economics, political science, public health, and medicine;

*Leadership in education, training, and public engagement*, spanning academic institutions, industry, policymaking, and society at large.

Such engagement enables statistical science to shape a wide range of domains beyond Statistics itself. To some extent, this transfer can also occur from the "backyard" bringing foundational principles for design of experiments, data analysis and correct solid interpretation. However, participation in the "frontyard" additionally allows the discipline to shape the broader scientific and societal environment through a visible intellectual culture. The contribution is then not limited to statistical methods and models alone, but extends to the implicit modes of reasoning and decision-making brought by people trained in Statistics. Statisticians in leadership positions within

---

[4]At the individual level, this is often a matter of personal scientific style.

companies, research institutions, or interdisciplinary teams naturally bring "statistical thinking" into discussions and problem-solving processes. This applies not only to highly visible leaders, but also to all individuals who take responsibility for advancing collaborative work.

Developing deep methodological innovations that elevate Statistics to a prominent position across science and engineering is increasingly difficult, especially in the current era of machine learning and AI. Yet when at least some statisticians assume leadership roles in large interdisciplinary efforts, whether in academia or industry, the discipline becomes more visible, and its distinctive strengths, particularly its broad interdisciplinarity and its emphasis on rigorous uncertainty quantification, can exert influence on a larger scale. In my subjective observation, statisticians often prefer to work independently on intellectually appealing problems while remaining comparatively modest in public visibility. Although this attitude is understandable, it does little to realize the discipline's broader potential to participate more actively in the "frontyard" – a development that, in my view, is increasingly necessary in current and future times.

## 3.2 AI and Foundation Models

AI and Foundation Models are likely to continue developing rapidly, and Statistics will need to define and strengthen its role within this major emerging area. Foundation Models perform a form of intelligent data compression, yielding powerful and broadly reusable representations. This enables easy access to information extracted from massively large-scale data, but the compression is necessarily lossy. Statistical methods then provide principled ways to adapt and assess these representations for specific domains and scientific questions.

### 3.2.1 An example of Statistics in AI: Evaluating Virtual Cell Perturbation Models

A concrete example of a statistical challenge arising in the context of AI and foundation models concerns the design of meaningful performance metrics and the proper evaluation of predicted perturbation effects (i.e., treatments) in high-throughput genomics and proteomics. I intentionally used the term "virtual cell perturbation models" in the section title, adopting the language currently popular in modern computational biology. This example, which relates to some of my recent research interests, illustrates how fundamental statistical questions re-emerge in contemporary large-scale predictive settings.

Predicting the effects of gene knock-outs on thousands of gene expression measurements, or forecasting the impact of drugs on high-dimensional protein expression profiles, is currently an active area of research. The latter problem is closer to clinical application, whereas gene knock-outs are experimentally more accessible and easier to measure at scale.

Modern AI approaches, in particular large deep-learning models and foundation models, have recently been proposed to substantially improve the prediction of unseen gene perturbations (see, e.g., Roohani et al., 2024). A lively debate followed. Ahlmann-Eltze, Huber, and Anders argued that deep-learning approaches do not yet outperform simple additive baselines (Ahlmann-Eltze et al., 2025). Shortly thereafter, Miller et al. responded that deep-learning models *do* outperform uninformative baselines when evaluated using well-calibrated metrics (Miller et al., 2025). [5]

This debate exposes a deeper statistical question: how should predictive performance be measured when the response is high-dimensional, often comprising thousands of correlated outcomes? A common but naive choice is the squared Euclidean norm, $\|\hat{Y} - Y\|_2^2$. However, from classical statistical theory we know that it can behave poorly in high dimensions and may lead to misleading conclusions: issues that are conceptually related to Stein's paradox (Stein, 1956). Careful metric calibration and meaningful performance measurement are therefore essential. Sparsifying the metric by concentrating weights on fewer high-dimensional components can markedly improve signal detection. Moreover, because multiple single cells are predicted, the predictions and targets form empirical distributions: Proper Scoring Rules (Gneiting and Raftery, 2007) have vastly improved probabilistic forecast evaluation and are equally valuable in this context. Such and related statistical ideas have recently been developed in this context (BLINDED, 2026; Nicol et al., 2026). When shifting from gene knock-outs to drug treatments and, ultimately, to predicting phenotypic outcomes, further challenges arise. Of particular interest is the prediction of previously unseen drug *combinations* on proteomics from single drug perturbations (see, e.g.,

---

[5]The references cited here are illustrative rather than exhaustive.

Sun et al., 2025), and the assessment of their efficacy. Here, the question is not merely predictive accuracy but the reliable identification of biologically and therapeutically actionable effects.

This example illustrates a broader point: even in domains where AI and large foundation models appear to dominate, fundamental statistical questions about evaluation, calibration, decision alignment, and generalization under distributional shift remain central. Addressing them is not peripheral but central to ensuring that advances in predictive modeling translate into scientifically credible and clinically useful conclusions.

## 3.3 Positioning

Statistics should position itself proactively within a highly competitive landscape in which machine learning and AI are currently attracting enormous attention.

**Playing in the frontyard.** This connects directly to Section 3.1.1, where I emphasized the importance of Statistics taking visible co-leadership roles in major activities across science, education, and institutional leadership. The long-term influence of the discipline will depend not only on methodological innovation but also on helping define interdisciplinary research agendas, educational programs, and the broader scientific culture.

**Maintaining core statistical territory.** Certain domains remain firmly grounded in statistical methodology. A primary example is clinical research, where clinical trials, study design, and drug approval processes rely fundamentally on statistical principles. It stands out as a particularly important area in which statistical approaches remain highly relevant and often dominant. Other application domains are exemplified in Section 3.1.1. Whenever rigorous standards for reliability, uncertainty quantification, and validation are required, such areas provide substantial long-term opportunities for the continued development and impact of Statistics.

**Blurred disciplinary boundaries.** In some areas, the boundaries between Statistics and Machine Learning will continue to blur. This cross-fertilization is intellectually enriching but may also create challenges concerning disciplinary identity, funding structures, and evaluation criteria. Statistics should therefore remain open-minded; particularly in academic promotion and in defining standards of research excellence.

At the same time, the field should remain attractive to researchers from neighboring areas, including Machine Learning, as well as to scientists across many disciplines. Maintaining a high level of credibility and methodological rigor is one of the most valuable assets of Statistics and should remain a central priority.

More broadly, diversity of scientific cultures is essential for healthy intellectual progress. If scientific development were driven predominantly by a single culture – such as that of Computer Science and Machine Learning – both research and education could risk becoming too narrowly focused. Statistics, together with other disciplines, helps maintain this diversity of perspectives and will therefore remain an essential component of the broader scientific ecosystem. The blurring of disciplinary boundaries might seem to conflict with a visible "frontyard" role (Section 3.1.1). Yet interdisciplinarity requires distinct fields to exist, thriving on the interaction of different scientific cultures. Statistics must therefore remain visible in the frontyard, standing alongside other disciplines as an essential contributor to a diverse scientific ecosystem.

**Education.** Education is generally among the most valuable strategic assets of modern societies. Industrial companies, public institutions, and research organizations alike depend on highly trained individuals with deep analytical expertise. A rigorous education forms the foundation of sustainable know-how and long-term innovation capacity.

In this context, a strong, visible, and forward-looking education in Statistics is of central importance. This is particularly true in an era increasingly shaped by Artificial Intelligence and automated decision systems. While AI technologies permeate many sectors of the economy and society, they do not replace statistical thinking nor the process of (mathematical) modeling. The ability to formulate domain problems in precise mathematical terms, to construct and analyze models, to develop sound methodology and efficient algorithms, and to critically validate assumptions, results, and conclusions remains indispensable. Statistical education should therefore rest on three interdependent pillars:

*Foundational thinking.* A solid grounding in mathematics and principled model formulation. This foundation enables abstraction, rigor, and the transfer of methods across domains.

*Computational and methodological integration, including AI tools.* The ability to use modern computational tools, including Machine Learning and AI techniques, and to apply them responsibly to complex problems arising in science, industry, and society.

*Statistical literacy*: Interpretation and communication. Competence in drawing valid conclusions under uncertainty, assessing limitations, ensuring reproducibility, and communicating results clearly and transparently to interdisciplinary collaborators, decision-makers, and the broader public. This pillar is as essential as the other two, and its value often becomes particularly apparent when statisticians work outside academia.

Only by integrating these three pillars can statistical education prepare students to critically use and evaluate emerging technologies, and assume responsibility in data-driven decision processes.

## 3.4   On the use of AI tools

My final thoughts concern broader developments that extend well beyond Statistics itself. The use of AI tools has already substantially accelerated the intellectual search for knowledge. Progress on open problems in mathematics is now reported at remarkable speed. For example, in combinatorics, a major open problem was recently solved through a counterexample generated by GPT-5 (Alon et al., 2026). While this achievement is impressive, generating a counterexample may be easier than establishing a positive result through a constructive proof. In mathematical statistics, recent work on density estimation for mixture models benefited from AI-assisted literature search and connections to related problems (Bubeck et al., 2025, Sec. II.1), while a minimax-optimal error rate was derived with significant use of GPT-5 (Dobriban, 2025).

Already today, and even more so in the near future, researchers have access to increasingly powerful technical assistance for mathematical derivations, the generation of proof strategies, literature synthesis, and accelerated programming. At the same time, all such outputs require careful validation and critical assessment.

**The importance of Problem Framing.**   The creative task of identifying and formulating meaningful problems – what I refer to as "problem framing" – will become even more important. Statistics has a long tradition of engaging deeply with scientific questions and translating them into well-defined analytical problems. Building on this tradition, the discipline is well positioned to emphasize and further develop the art of problem framing in the age of AI.

**Simplicity, conceptual clarity and elegance.**   Simplicity is essential for communication and understanding – and ultimately for impact in science, industrial governance, and politics. Statistics is well positioned to decompose complex problems into understandable components. More broadly, any discipline based on mathematical modeling simplifies reality in order to enable clearer reasoning and, potentially, better decisions. George Box stated, "All models are wrong, but some are useful." One should add: "Models are useful because they provide a basis for clear communication." In an era of increasingly automated and opaque systems, Statistics should preserve and strengthen its commitment to simplicity, interpretability, and conceptual clarity.

Elegance is closely related to beauty and often contains an element of surprise. The elegance of a derivation, an argument, or the design of a methodology may be partly subjective, but it is important. It is closely connected to creativity, which remains a deeply human endeavor. The art of science is a beautiful act of creation – and it will continue to be valued by many.

## 4   Conclusion

The developments of the past twenty-five years demonstrate that Statistics continues to generate conceptual breakthroughs with broad scientific impact. The field now interacts more closely than ever with Machine Learning and Artificial Intelligence. Despite this rapidly evolving landscape, Statistics retains a clear identity grounded in interdisciplinary motivation, rigorous methodology, principled modeling, and a strong commitment to reliability and replicability.

Statistics should therefore continue to engage actively with emerging computational and AI-based technologies. However, this development also calls for principles and standards that still need to be more clearly established. These include safeguarding the core values of reliability, transparency, and reproducibility, all of which

are deeply connected to the mathematical foundations of Statistics. They also include a broader sense of professional responsibility that extends beyond methodological caution to encompass the consequences of decisions informed by AI systems, as well as wider ethical considerations. Sustaining these strengths will require strong and forward-looking educational programs.

A major issue that must be addressed concerns the following: who will actually have access to such educational programs in Statistics? It cannot be taken for granted that broad groups of students, scientists, or professionals will engage with statistical education, even if it is thoughtfully modernized. Likewise, whose problems receive the careful analytical attention advocated by Statistics? These questions are closely connected to the idea of "Statistics in the frontyard" (Section 3.1.1). To remain influential and socially relevant, Statistics must engage visibly in educational development and in major initiatives across science, industry, government, and society. Statistical societies, as collective communities of hundreds or thousands of researchers and educators, can play an eminent role in shaping and advancing these efforts.

Ultimately, the value of Statistics lies not only in algorithms or AI technologies but in a broader intellectual approach: the ability to formulate meaningful problems, reason under uncertainty, and translate complex data into reliable scientific knowledge. In an era increasingly shaped by automated systems and large-scale predictive models, maintaining diversity in scientific perspectives and methodological cultures will be essential. Statistics – together with other disciplines – will remain indispensable for ensuring credible and responsible scientific progress.

**Acknowledgements.** I thank an anonymous reviewer for highly insightful and constructive comments that substantially improved the manuscript, and BLINDED for feedback on using AI in mathematics. The author used ChatGPT (OpenAI, GPT-5) for language editing and takes full responsibility for the content.

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
