# OpenReview forum: "Statistics in the Next Quarter-Century: Playing also  in the Frontyard?"
_SLADS/Section_A — Accepted by SLADS_Section_A_

### Review · Reviewer_kpZY · 2026-05-25

**Summary Of Contributions:**

This is a perspective piece whose main contribution is a coherent argument that Statistics retains a distinctive and defensible identity within the AI and Data Science landscape. The survey of the past 25 years of conceptual innovation is a useful synthesis, the Tukey backyard framing names a tension the field feels but rarely articulates directly, and the appendix makes a genuine technical point about evaluation metric design in high-dimensional settings that connects current AI debates to classical statistical theory. The overall case, where reliability, uncertainty quantification, and principled inference are not replicable by AI methods alone, is worth making, and this paper makes a reasonable start at it.

**Audience:**

Yes

**Broader Impact Concerns:**

Neither concern is serious enough to require a formal Broader Impact Statement. The paper would benefit from briefly acknowledging two understated tensions: that encouraging statisticians to adopt AI tools in high-stakes applied settings carries professional responsibility implications that go beyond methodological caution, and that the case for Statistics as a societal asset sits awkwardly alongside the paper's silence on who has access to rigorous statistical training and whose problems tend to get the careful analytical attention the paper advocates for. A sentence or two on each in the conclusion would be sufficient.

**Claims And Evidence:**

Yes

**Requested Changes:**

1. The appendix case study should be moved into the main body and expanded to make explicit what is actually at stake when evaluation metrics are poorly chosen in high-dimensional predictive settings.
2. The paper's repeated claim that Statistics is "rooted in applications" needs to be substantiated in the main text through specific engagement with at least two or three applied domains beyond clinical research, with enough detail to show what is lost when statistical rigor is displaced.
3. Section 2.2 raises the question of whether Statistics is structurally confined to a supporting role and then abandons it, and the revision must either work through this with concrete examples of applied statistical leadership or argue explicitly why the tension is unresolvable and what that implies for the field.
4. The third educational pillar, i.e., communication and validation of uncertainty, should be developed with the same seriousness as the other two, given that it is the skill most visibly tested when statisticians work outside academia.

**Strengths And Weaknesses:**

Strengths:
1. The choice of topic and the author's willingness to be self-critical about the discipline;
2. Section 2.3's survey of innovations: the citation comparisons (PRISMA vs. Krizhevsky) are called out specifically as effective rhetorical moves;
3. The Tukey metaphor for naming a tension that statisticians feel but rarely say out loud;
4. The appendix as paradoxically the paper's best section: the Stein's paradox connection and the proper scoring rules suggestion are flagged as genuine contributions;
5. The education framework is a serious attempt rather than a placeholder.

Weaknesses:
1. The mismatch between "rooted in applications" and a paper that's mostly disciplinary history;
2. Narrow sectoral coverage (finance, climate, policy, tech experimentation all missing);
3. The education pillar on communication is the most important and least developed
4. The backyard question was left unresolved, with no practical examples of statistics leading

---

> ### Author Response · Authors · 2026-06-01
>
> Thank you very much for your most insightful comments, which inspired me to substantially revise some parts of the paper. The main changes in the revised manuscript are indicaed in red color.
>
> Regarding your requested changes:
>
> **Re 1:** The appendix has been moved to the main body and now appears as a new Subsection 3.2.1. I have embedded it within a slightly expanded Section 3.2.
>
> **Re 2:** I have added a few more applied domains in Section 2.2 (normalization and data processing in genomics) and in Section 3.1.1. The second-to-last paragraph of Section 3.1.1 also implicitly addresses your point
>
> > "with enough detail to show what is lost when statistical rigor is displaced."
>
> However, I have emphasized mainly what is gained rather than what is lost. Also, I think it is more appropriate to point to the general gains rather than to technical details.
>
> **Re 3:** I have added my own perspective on the question of "statistics as a helping science" and introduced a new Subsection 3.1.1 on *Statistics in the frontyard*. This part is really central and, consequently, I have also changed the title of the paper so that "frontyard" now appears in the title. The concept of the frontyard also appears in Section 3.3 (Positioning) and Section 4 (Conclusion).
>
> **Re 4:** I added a remark on the equal importance of the third educational pillar.
>
> **Re Broader Impact Concerns:** I added sentences in Section 4 (Conclusion) concerning
>
> > "encouraging statisticians to adopt AI tools in high-stakes applied settings carries professional responsibility implications that go beyond methodological caution"
>
> and
>
> > "the case for Statistics as a societal asset sits awkwardly alongside the paper's silence on who has access to rigorous statistical training and whose problems tend to get the careful analytical attention the paper advocates for."

---

> > ### Comment · Reviewer_kpZY · 2026-06-09
> >
> > Thank you for your efforts on revising the manuscript. All my comments are well addressed.

---

### Decision · Action_Editor_X1Ww · 2026-06-09

**Recommendation:** Accept as is

**Audience:**

I believe the findings and perspectives presented in this paper will be of significant interest to a large portion of the SLADS udience. As a journal dedicated to advancing the frontiers of statistics, machine learning, and data science, the SLADS readership is highly invested in discussions regarding disciplinary identity, cross-fertilization, and long-term research directions. The manuscript directly addresses the evolving ecosystem of data sciences and the conceptual overlaps between traditional statistical workflows and large-scale AI frameworks.

The paper bridges the gap between different methodological communities. It provides a forward-looking roadmap that will stimulate meaningful discourse among statisticians, ML practitioners, and data scientists alike, making it highly relevant to the journal's audience.

**Claims And Evidence:**

The claims made in this submission are well-supported by convincing, and clear evidence appropriate for a perspective paper. The author centralizes the argument that Statistics must actively engage as a co-leading discipline in the "frontyard" of major interdisciplinary endeavors rather than remaining solely a validation-oriented "helping science" in the backyard.

To back these assertions, the manuscript integrates a multi-faceted layer of evidence: quantitative labor market trends from the U.S. Bureau of Labor Statistics (showcasing the rapid growth of data-oriented professions), a thorough retrospective of foundational statistical breakthroughs over the past 25 years (such as FDR, high-dimensional sparsity, conformal inference, and the open-source R project), and a concrete, contemporary case study on evaluating virtual cell perturbation foundation models. The discussion surrounding high-dimensional predictive performance metrics and its connection to classical concepts like Stein's paradox provides clear evidence of how principled statistical reasoning remains indispensable in the modern AI era.